# Regulation of Epithelial–Mesenchymal Transition Pathway and Artificial Intelligence-Based Modeling for Pathway Activity Prediction

**Shihori Tanabe** [1,*] **, Sabina Quader** [2] **, Ryuichi Ono** [3] **, Horacio Cabral** [4] **, Kazuhiko Aoyagi** [5] **, Akihiko Hirose** [1] **, Edward J. Perkins** [6] **, Hiroshi Yokozaki** [7] **and Hiroki Sasaki** [8]

1   Division of Risk Assessment, Center for Biological Safety and Research, National Institute of Health Sciences, Kawasaki 210-9501, Japan
2   Innovation Center of NanoMedicine (iCONM), Kawasaki Institute of Industrial Promotion, Kawasaki 210-0821, Japan
3   Division of Cellular and Molecular Toxicology, Center for Biological Safety and Research, National Institute of Health Sciences, Kawasaki 210-9501, Japan
4   Department of Bioengineering, Graduate School of Engineering, The University of Tokyo, Tokyo 113-0033, Japan
5   Department of Clinical Genomics, National Cancer Center Research Institute, Tokyo 104-0045, Japan
6   Environmental Laboratory, US Army Engineer Research and Development Center, Vicksburg, MS 39180, USA
7   Department of Pathology, Kobe University of Graduate School of Medicine, Kobe 650-0017, Japan
8   Department of Translational Oncology, National Cancer Center Research Institute, Tokyo 104-0045, Japan
*   Correspondence: stanabe@nihs.go.jp; Tel.: +81-44-270-6686

**Simple Summary:** Molecular network pathways are activated or inactivated under various conditions. Previously, we revealed that epithelial–mesenchymal transition (EMT) is a feature of diffuse-type gastric cancer. Here, we modeled the activation states of EMT in the development pathway using molecular pathway images and artificial intelligence (AI). The regulation of EMT in the development pathway was activated in diffuse-type gastric cancer (GC) and inactivated in intestinal-type GC. AI modeling with molecular pathway images generated a highly accurate Elastic-Net Classifier models that was validated with 10 additional activated and 10 inactivated pathway images.

**Abstract:** Because activity of the epithelial–mesenchymal transition (EMT) is involved in anti-cancer drug resistance, cancer malignancy, and shares some characteristics with cancer stem cells (CSCs), we used artificial intelligence (AI) modeling to identify the cancer-related activity of the EMT-related pathway in datasets of gene expression. We generated images of gene expression overlayed onto molecular pathways with Ingenuity Pathway Analysis (IPA). A dataset of 50 activated and 50 inactivated pathway images of EMT regulation in the development pathway was then modeled by the DataRobot Automated Machine Learning platform. The most accurate models were based on the Elastic-Net Classifier algorithm. The model was validated with 10 additional activated and 10 additional inactivated pathway images. The generated models had false-positive and false-negative results. These images had significant features of opposite labels, and the original data were related to Parkinson's disease. This approach reliably identified cancer phenotypes and treatments where EMT regulation in the development pathway was activated or inactivated thereby identifying conditions where therapeutics might be applied or developed. As there are a wide variety of cancer phenotypes and CSC targets that provide novel insights into the mechanism of CSCs' drug resistance and cancer metastasis, our approach holds promise for modeling and simulating cellular phenotype transition, as well as predicting molecular-induced responses.

**Keywords:** artificial intelligence; epithelial–mesenchymal transition; Ingenuity Pathway Analysis; machine learning; molecular pathway network

## 1. Introduction

Molecular network pathways are activated or inactivated under many different conditions. Previously, we found that diffuse-type gastric cancer (GC) has a feature of epithelial–mesenchymal transition (EMT) [1–3]. EMT is involved in anti-cancer drug resistance, cancer malignancy, metastasis, and cancer stem cells (CSCs) [4–7]. Experiments in anti-cancer drug-resistant cancer cell lines indicate that EMT is involved in cancer cell drug resistance [8], highlighting the significance of EMT targeting in cancer treatment [6].

Several signaling pathways involved in EMT contribute to drug resistance [6]. Tumor growth factor beta (TGFβ) signaling activates SMAD2/3, which then complexes with SMAD4 to form a trimetric SMAD complex, leading to the transcription of EMT transcription factors [9]. Wnt/β-catenin signaling activates Snail transcription to induce EMT [6,10]. Recent studies have also revealed the role of EMT in autophagy and CSCs during metastasis [11,12]. However, the relationship between the EMT pathway activation state and therapeutic responsiveness is not fully understood.

Understanding the activity state of the EMT pathway in cancer cells may be an important clue for identifying therapeutic targets in malignant cancers. To effectively predict EMT activity and potential therapeutic responsiveness, molecular pathway images were used to capture activity of EMT-related pathways of datasets in Ingenuity Pathway Analysis (IPA), followed by artificial intelligence (AI) modeling with images of gene expression activity in the pathway.

## 2. Materials and Methods

### 2.1. Data Analysis of Diffuse- and Intestinal-Type GC

We used RNA sequencing data of diffuse- and intestinal-type GC, which are publicly available in The Cancer Genome Atlas (TCGA) of the cBioPortal for Cancer Genomics database at the National Cancer Institute (NCI) Genomic Data Commons (GDC) data portal [13–17]. Publicly available data on stomach adenocarcinoma in the TCGA, Stomach Adenocarcinoma (TCGA, PanCancer Atlas), [13–16] were compared between diffuse-type GC, which is genomically stable (n = 50), and intestinal GC, which has a feature of chromosomal instability (n = 223), in TCGA Research Network publications, as previously described [1,14,18].

### 2.2. Network Analysis

Data on intestinal- and diffuse-type GC from the TCGA cBioPortal for Cancer Genomics were uploaded and analyzed using IPA (Qiagen, CA, USA) [19,20]. The datasets of gene expression in diseases were searched in IPA, and datasets with absolute values in z-score in the top 60 for activated state and inactivated state (total of 120) in regulation of EMT in the development pathway were extracted for AI prediction modeling and evaluation. Among 120 analyses in the activity plot of regulation of EMT in the development pathway, 50 activated and 50 inactivated analyses (total of 100) were used to generate AI models and 10 activated and 10 inactivated analyses (total of 20) were withheld for use in validating the generated model. The 100 analyses (50 activated and 50 inactivated states) found in the database of IPA and newly used to generate AI-based models are summarized in Table 1.

**Table 1.** Analyses in the regulation of EMT in the development pathway for AI prediction modeling.

| Analysis Name | Disease State | Target Gene | Treatment | EMT |
|---|---|---|---|---|
| 996-Breast ductal carcinoma torin 2 28190 | Breast ductal carcinoma | Mtor | Torin 2 | TRUE |
| 16332-Fibrocystic breast disease neratinib 7038 | Fibrocystic breast disease | Her2; egfr | Neratinib | TRUE |
| 16885-Fibrocystic breast disease erlotinib 7651 | Fibrocystic breast disease | Egfr | Erlotinib | TRUE |
| 116-Bone osteosarcoma (OS) MK2206 2727 | Bone osteosarcoma (OS) | | MK2206 | TRUE |
| 1766-Breast ductal carcinoma brivanib 8512 | Breast ductal carcinoma | Vegfr; fgfr | Brivanib | TRUE |
| 47-Huntington's disease (HD) haloperidol 12804 | Huntington's disease (HD) | | Haloperidol | TRUE |
| 4874-Melanoma crizotinib 22540 | Melanoma | Alk and ros1 | Crizotinib | TRUE |
| 6785-Non-small cell lung carcinoma ZSTK474 24663 | Non-small cell lung carcinoma | PI3K | ZSTK474 | TRUE |
| 7-Normal control differentiation medium 10230 | Normal control | | Differentiation medium | TRUE |
| 13972-Prostate adenocarcinoma (PRAD) PI103 4415 | Prostate adenocarcinoma (PRAD) | PI3K | PI103 | TRUE |
| 16046-Prostate adenocarcinoma (PRAD) MK2206 6720 | Prostate adenocarcinoma (PRAD) | AKT | MK2206 | TRUE |
| 7063-Breast adenocarcinoma linifanib 24973 | Breast adenocarcinoma | Rtk; vegf; pdgf | Linifanib | TRUE |
| 7923-Breast adenocarcinoma PF3758309 25928 | Breast adenocarcinoma | PAK4 | PF3758309 | TRUE |
| 2-Breast carcinoma beta-estradiol (E2) 3915 | Breast carcinoma | | B-estradiol (E2) | TRUE |
| 10974-Breast ductal carcinoma KIN001-043 1084 | Breast ductal carcinoma | GSK3β | KIN001-043 | TRUE |
| 1116-Breast ductal carcinoma QL-X-138 1291 | Breast ductal carcinoma | BTK; MNK | QL-X-138 | TRUE |
| 29-Colon cancer GSK525762A; trametinib 3009 | Colon cancer | | GSK525762A; trametinib | TRUE |
| 35-Colon cancer active JQ1 1658 | Colon cancer | | Active JQ1 | TRUE |
| 13176-Colorectal adenocarcinoma BGJ398 3531 | Colorectal adenocarcinoma | FGFR | BGJ398 | TRUE |
| 12948-Colorectal adenocarcinoma AZ628 3277 | Colorectal adenocarcinoma | BRAF; BRAFV600E; C-RAF-1 | AZ628 | TRUE |
| 12715-Colorectal adenocarcinoma AT7519 3019 | Colorectal adenocarcinoma | CDK | AT7519 | TRUE |
| 6-Disease control IL-1 beta 15814 | Disease control | | IL-1β | TRUE |
| 17104-Fibrocystic breast disease canertinib 7896 | Fibrocystic breast disease | Egfr; her2; erbb4 | Canertinib | TRUE |
| 17239-Fibrocystic breast disease torin 1 8045 | Fibrocystic breast disease | Mtor | Torin 1 | TRUE |
| 16449-Fibrocystic breast disease AZD8330 7167 | Fibrocystic breast disease | MEK | AZD8330 | TRUE |
| 17590-Fibrocystic breast disease mitoxantrone 8435 | Fibrocystic breast disease | Topoisomerase | Mitoxantrone | TRUE |
| 7-Fibrosis DMSO 7394 | Fibrosis | | DMSO | TRUE |
| 20894-Hepatocellular carcinoma (LIHC) chelerythrine chloride 12106 | Hepatocellular carcinoma (LIHC) | PKC | Chelerythrine chloride | TRUE |
| 59-Huntington's disease (HD) nortriptyline 12817 | Huntington's disease (HD) | | Nortriptyline | TRUE |
| 2-Lung adenocarcinoma (LUAD) Transfection_HOXC6 631 | Lung adenocarcinoma (LUAD) | | Transfection_HOXC6 | TRUE |
| 3-Major depressive disorder differentiation medium 3130 | Major depressive disorder | | Differentiation medium | TRUE |
| 5612-Melanoma AT7867 23361 | Melanoma | AKT1/2/3; p70s6k/PKA | AT7867 | TRUE |
| 5173-Melanoma lapatinib 22873 | Melanoma | Her2; egfr | Lapatinib | TRUE |
| 91-Non-small cell lung carcinoma BGT226 27235 | Non-small cell lung carcinoma | PI3K; mtor | BGT226 | TRUE |
| 14456-Normal control WYE125132 4953 | Normal control | Mtor | WYE125132 | TRUE |
| 28175-Normal control glesatinib 20196 | Normal control | C-met; tek; vegfr; ron | Glesatinib | TRUE |
| 60-Normal control 567 | Normal control | | | TRUE |
| 2-Normal control culture medium 1187 | Normal control | | Culture medium | TRUE |

**Table 1.** *Cont.*

| Analysis Name | Disease State | Target Gene | Treatment | EMT |
|---|---|---|---|---|
| 9914-Normal control EX527 28140 | Normal control | SIRT1 | EX527 | TRUE |
| 4-Normal control suberoylanilide hydroxamic acid (SAHA) 2204 | Normal control | | Suberoylanilide hydroxamic acid (SAHA) | TRUE |
| 27560-Normal control BMS509744 19513 | Normal control | ITK | BMS509744 | TRUE |
| 14256-Normal control AZD8055 4731 | Normal control | Mtor | AZD8055 | TRUE |
| 19-Normal control no serum 3447 | Normal control | | No serum | TRUE |
| 5-Parkinson's disease (PD) differentiation medium 4389 | Parkinson's disease (PD) | | Differentiation medium | TRUE |
| 23661-Prostate adenocarcinoma (PRAD) AZD5438 15181 | Prostate adenocarcinoma (PRAD) | CDK | AZD5438 | TRUE |
| 25661-Breast adenocarcinoma omipalisib 17403 | Breast adenocarcinoma | Pi3k | Omipalisib | TRUE |
| 90-Prostate adenocarcinoma (PRAD) monolayer culture 4346 | Prostate adenocarcinoma (PRAD) | | Monolayer culture | TRUE |
| 8-Normal control lipopolysaccharide (LPS) 4907 | Normal control | | Lipopolysaccharide (LPS) | TRUE |
| 2-Acute myeloid leukemia (LAML) lipopolysaccharide (LPS) 9357 | Acute myeloid leukemia (LAML) | | Lipopolysaccharide (LPS) | TRUE |
| 25084-Breast adenocarcinoma CGP60474 16762 | Breast adenocarcinoma | CDK1; CDK2 | CGP60474 | TRUE |
| 20-Non-small cell lung carcinoma IFN gamma 13421 | Non-small cell lung carcinoma | | Ifnγ | FALSE |
| 7-Normal control co-culture 3087 | Normal control | | Co-culture | FALSE |
| 5-Normal control hypoxia 13911 | Normal control | | Hypoxia | FALSE |
| 1-Normal control IFN alpha 4636 | Normal control | | Ifnα | FALSE |
| 11-Normal control differentiation medium 10205 | Normal control | | Differentiation medium | FALSE |
| 3-Normal control Infection_human betaherpesvirus 5 (HHV5) 15858 | Normal control | | Infection_human betaherpesvirus 5 (HHV5) | FALSE |
| 31-Bone osteosarcoma (OS) 1,9-pyrazoloanthrone 2804 | Bone osteosarcoma (OS) | | 1,9-pyrazoloanthrone | FALSE |
| 57-Coronavirus disease 2019 (COVID-19) 96 | Coronavirus disease 2019 (COVID-19) | | | FALSE |
| 17503-Fibrocystic breast disease HG6-64-1 8339 | Fibrocystic breast disease | B-RAF | HG6-64-1 | FALSE |
| 11-Genetic disease 444 | Genetic disease | | | FALSE |
| 4-Glioblastoma (GBM) differentiation medium 6303 | Glioblastoma (GBM) | | Differentiation medium | FALSE |
| 23448-Hepatocellular carcinoma (LIHC) imatinib 14944 | Hepatocellular carcinoma (LIHC) | BCR-ABL | Imatinib | FALSE |
| 86-Huntington's disease (HD) sodium butyrate 12847 | Huntington's disease (HD) | | Sodium butyrate | FALSE |
| 21-Mantle cell lymphoma DMSO 3032 | Mantle cell lymphoma | | DMSO | FALSE |
| 5-Non-alcoholic steatohepatitis (NASH) none 11484 | Non-alcoholic steatohepatitis (NASH) | | None | FALSE |
| 10431-Normal control RAF265 482 | Normal control | C-RAF; B-RAF; B-RAFV600E | RAF265 | FALSE |
| 11-Normal control differentiation medium 4490 | Normal control | | Differentiation medium | FALSE |
| 14744-Normal control dasatinib 5273 | Normal control | Src family | Dasatinib | FALSE |
| 65-Normal control IL-3 17225 | Normal control | | IL-3 | FALSE |
| 14639-Normal control saracatinib 5156 | Normal control | Src; bcr-abl | Saracatinib | FALSE |
| 3-Normal control DHA-5-HT 4554 | Normal control | | DHA-5-HT | FALSE |

Table 1. *Cont.*

| Analysis Name | Disease State | Target Gene | Treatment | EMT |
|---|---|---|---|---|
| 28-Prostatic intraepithelial neoplasia (PIN) plumbagin 49 | Prostatic intraepithelial neoplasia (PIN) | | Plumbagin | FALSE |
| 4-Normal control differentiation medium 3415 | Normal control | | Differentiation medium | FALSE |
| 9-Huntington's disease (HD) meclizine 12851 | Huntington's disease (HD) | | Meclizine | FALSE |
| 6-Normal control culture medium 593 | Normal control | | Culture medium | FALSE |
| 22597-Normal control GSK429286A 13998 | Normal control | ROCK1; ROCK2 | GSK429286A | FALSE |
| 8-Normal control 3-D culture; co-culture; differentiation 3017 | Normal control | | 3D culture; co-culture; differentiation medium | FALSE |
| 110-Normal control 109 | Normal control | | | FALSE |
| 26-Bone osteosarcoma (OS) nilotinib 2798 | Bone osteosarcoma (OS) | | Nilotinib | FALSE |
| 26025-Breast adenocarcinoma saracatinib 17808 | Breast adenocarcinoma | Src; bcr-abl | Saracatinib | FALSE |
| 11577-Breast ductal carcinoma crizotinib 1754 | Breast ductal carcinoma | Alk and ros1 | Crizotinib | FALSE |
| 17316-Fibrocystic breast disease KIN001-043 8131 | Fibrocystic breast disease | GSK3β | KIN001-043 | FALSE |
| 2-Fibrosis SB525334 7389 | Fibrosis | | SB525334 | FALSE |
| 52-Huntington's disease (HD) meclizine 12810 | Huntington's disease (HD) | | Meclizine | FALSE |
| 1-Normal control culture medium 1186 | Normal control | | Culture medium | FALSE |
| 17-Normal control differentiation medium 4496 | Normal control | | Differentiation medium | FALSE |
| 6-Normal control hypoxia 13912 | Normal control | | Hypoxia | FALSE |
| 2-Major depressive disorder differentiation medium 3129 | Major depressive disorder | | Differentiation medium | FALSE |
| 11-Disease control none 4051 | Disease control | | None | FALSE |
| 10-Normal control 3-D culture; co-culture; differentiation 2995 | Normal control | | 3D culture; co-culture; differentiation medium | FALSE |
| 5-Normal control lipopolysaccharide (LPS) 15704 | Normal control | | Lipopolysaccharide (LPS) | FALSE |
| 1-Normal control differentiation medium 1246 | Normal control | | Differentiation medium | FALSE |
| 6-Normal control 151 | Normal control | | 3d culture; none | FALSE |
| 10-Normal control differentiation medium 4489 | Normal control | | Differentiation medium | FALSE |
| 13-Normal control co-culture 3079 | Normal control | | Co-culture | FALSE |
| 13051-Colorectal adenocarcinoma BMS777607 3393 | Colorectal adenocarcinoma | C-MET; AXL; RON; TYRO3 | BMS777607 | FALSE |
| 27-Huntington's disease (HD) meclizine 12782 | Huntington's disease (HD) | | Meclizine | FALSE |
| 8-Normal control GW3965 10098 | Normal control | | GW3965 | FALSE |
| 11-Normal control 368 | Normal control | | | FALSE |
| 6-Normal control culture medium 1191 | Normal control | | Culture medium | FALSE |

### 2.3. AI Prediction Modeling

To create a prediction model using multi-modal data including images and text descriptions of molecular networks, an enterprise AI platform (DataRobot Automated Machine Learning version 7.2; DataRobot Inc. (Boston, MA, USA) was used. For the modeling, the 100 molecular networks on the regulation of EMT in the development pathway were collected and input as image data in the DataRobot (50 images in the activated state and 50 images in the inactivated state), which automatically created and tuned prediction

models using various machine-learning algorithms (e.g., eXtreme gradient-boosted trees, random forest, regularized regression such as Elastic Net, Neural Networks) [21–23]. Finally, the AI model with the highest predictive accuracy on DataRobot was identified, and various insights (such as Permutation Importance or Partial Dependence Plot) obtained from the model were reviewed. To calculate the accuracy of the model, 20 additional image data (10 images in the activated state and 10 images in the inactivated state) that were not used as training data for the AI model creation were added for validation.

### 2.4. Statistical Analysis

The RNA sequencing data on diffuse- and intestinal-type GC was analyzed via Student's *t*-test. The z-scores of intestinal- and diffuse-type GC samples were compared, and the difference was considered significant at $p < 0.00001$, following previous reports [1,18]. The activation z-score in each pathway was calculated in IPA to show the level of activation.

## 3. Results

### 3.1. Regulation of the EMT in Development Pathway in Diffuse- and Intestinal-Type GC

3.1.1. Gene Expression Mapping in Regulation of the EMT in the Development Pathway in Diffuse- and Intestinal-Type GC

Alterations in gene expression in diffuse- and intestinal-type GC was mapped to a canonical pathway, "Regulation of the EMT in development pathway" (Figure 1) based on the previous gene expression analysis results [1]. Red or green color indicates upregulated or downregulated genes, respectively. In the regulation of EMT in the development pathway, frizzled and adenomatous polyposis coli regulator of the WNT signaling pathway (APC) was upregulated, while SUFU negative regulator of hedgehog signaling (SUFU), pygopus family PHD finger 2 (PYGO2), and BRCA1 was downregulated in diffuse-type GC compared to intestinal-type GC. APC encodes a tumor suppressor protein that acts as an antagonist of the Wnt signaling pathway. APC is also involved in other processes, including cell migration and adhesion, transcriptional activation, and apoptosis. SUFU is associated with β-catenin binding, protein kinase binding, and transcription regulation.

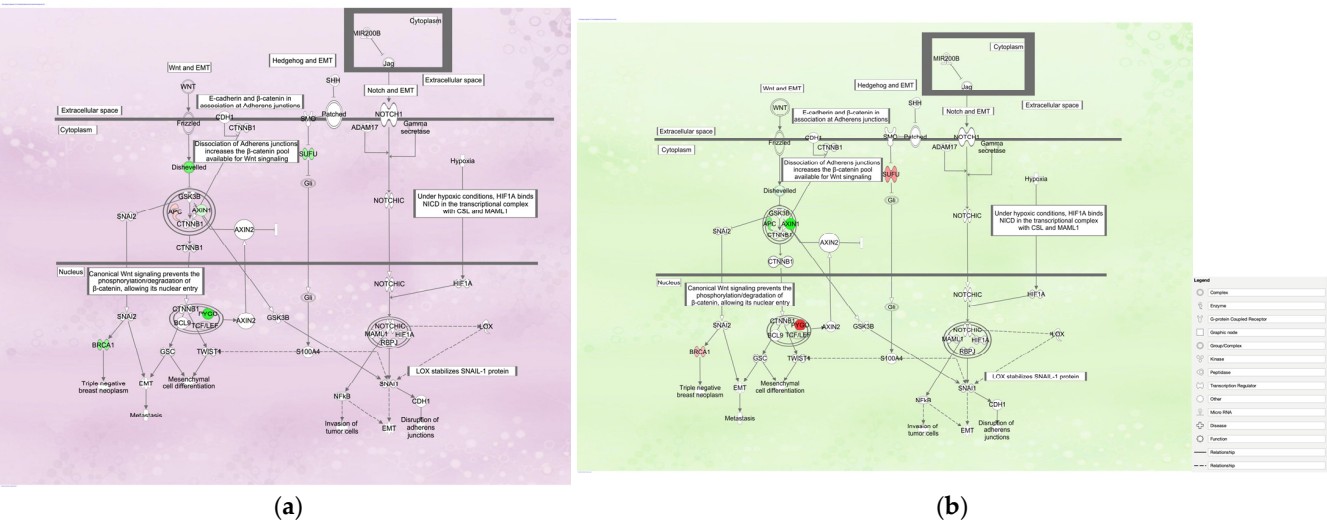

(**a**)                                                                                           (**b**)

**Figure 1.** Regulation of the epithelial–mesenchymal transition (EMT) in development pathway in diffuse- and intestinal-type gastric cancer (GC). (**a**) Gene expression alteration in diffuse-type GC in regulation of the EMT in development pathway; (**b**) Gene expression alteration in intestinal-type GC in regulation of the EMT in development pathway. Red or green color indicates upregulated or downregulated genes, respectively. The intensity of colors indicates the degree of up- or downregulation. A solid or dashed line indicates direct or indirect interaction, respectively.

### 3.1.2. Molecular Activity Prediction in Regulation of the EMT in Development Pathway in Diffuse- and Intestinal-Type GC

The prediction of molecular activity in the regulation of the EMT in the development pathway in diffuse- and intestinal-type GC was mapped (Figure 2). GSK3β, SNAI1, NFκB, LOX, and EMT are activated, whereas SNAI2 and E-cadherin are inactivated in diffuse-type GC compared to intestinal-type GC. Notch receptor 1 (NOTCH1) intracellular domain (NOTCHIC) was predicted to be activated in the CSL-HIF1A-MAML1-NICD complex, which consists of hypoxia-inducible factor 1 subunit alpha (HIF1A), mastermind-like transcriptional coactivator 1 (MAML1), NOTCH1, and recombination signal binding for immunoglobulin kappa J region (RBPJ) in the nucleus, and β-catenin (CTNNB1) was predicted to be activated in β-catenin-APC-AXIN-GSK3β complex in the cytoplasm in diffuse-type GC compared to intestinal-type GC.

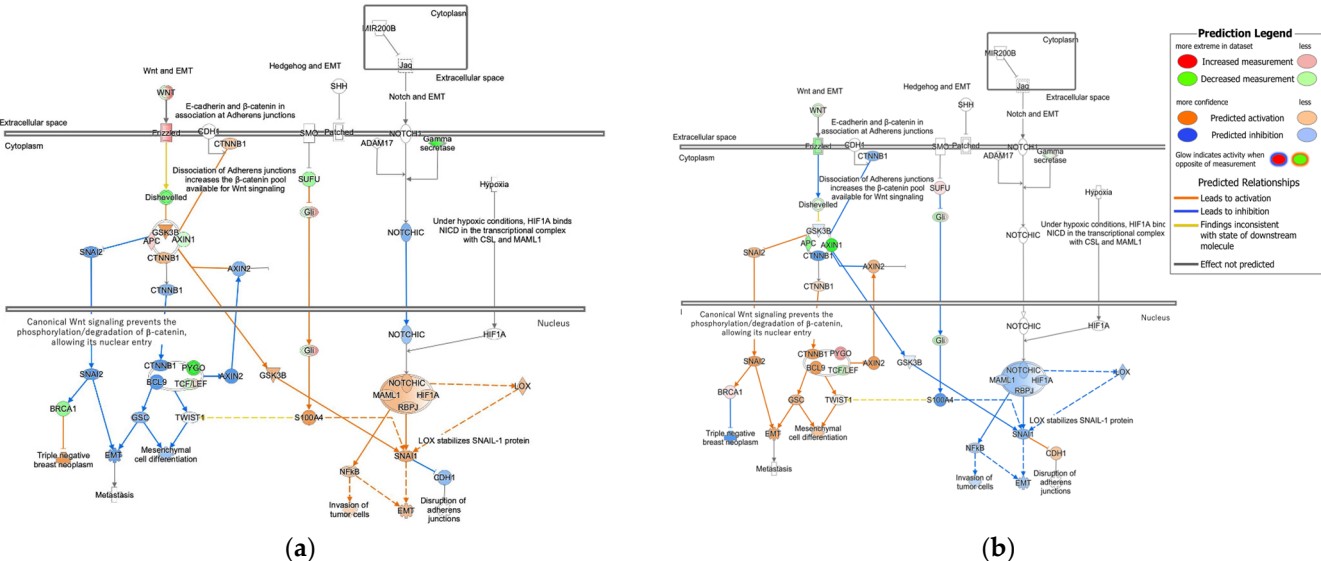

**Figure 2.** Molecular activity prediction in regulation of the EMT in development pathway in diffuse- and intestinal-type GC. (**a**) Molecular activity prediction in diffuse-type GC; (**b**) molecular activity prediction in intestinal-type GC. Red or green color indicates upregulated or downregulated genes, respectively. The intensity of colors indicates the degree of up- or downregulation. A solid or dashed line indicates direct or indirect interaction, respectively. Orange or blue color indicates predicted activation or inhibition, respectively. The intensity of colors indicates the confidence level of the prediction.

### 3.2. Activity Plot of Regulation of the EMT in Development Pathway

In total, 6216 analyses were found to be involved in the regulation of the EMT in the development pathway (as of September 2021) (Figure 3). In subsequent AI modeling analyses, samples with "NA" in the case treatment and blank in the disease state were excluded.

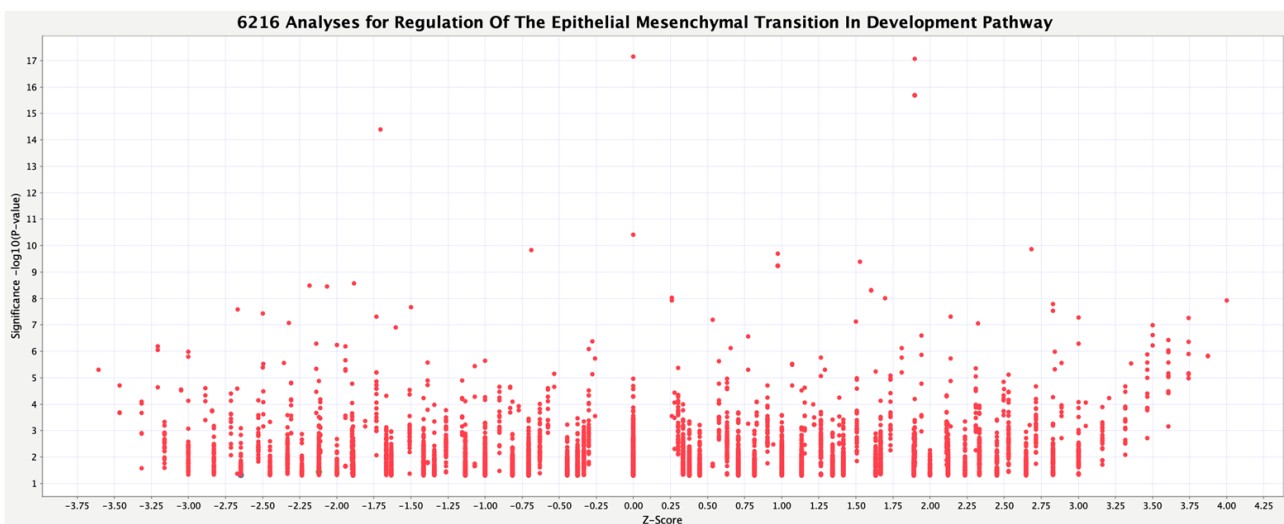

**Figure 3.** Activity plot of regulation of EMT in development pathway (6216 analyses, as of September 2021).

### 3.3. AI Modeling and Validation of the Prediction Model

The activation state of regulation of EMT in the development pathway was modeled by machine learning, including deep learning, using 50 activated and 50 inactivated images of the regulation of EMT in development pathway (Figure 4). DataRobot was used for machine-learning modeling and 34 models were automatically created, including an Elastic-Net Classifier (L2/Binomial Deviance) model. DataRobot also highlighted the parts of the image data critical to the prediction accuracy of the model in an activation map (Figure 4).

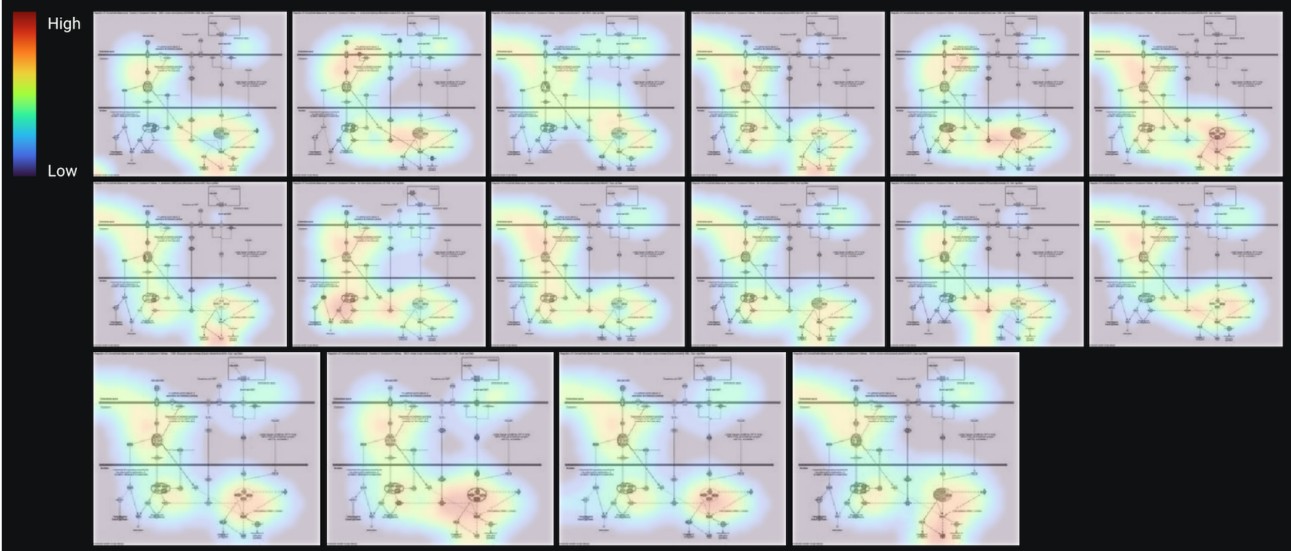

**Figure 4.** Activation map of AI modeling (DataRobot).

To validate the ElasticNet Classifier model, predictions were made using data on 10 activated and 10 inactivated pathway images that were not used to train the model (Table 2). The results showed that the prediction accuracy for the additional 20 images was 100% (AUC = 1.0).

**Table 2.** Validation of the model ElasticNet_Classifier_(L2/Binomial_Deviance).

| Analysis Name | Disease State | Target Gene | Tissue | Treatment | EMT | Prediction | Label |
|---|---|---|---|---|---|---|---|
| 18092-Breast adenocarcinoma CP466722 8993 | breast adeno-carcinoma | ATM | Breast | Cp466722 | TRUE | 0.9693884 | 1 |
| 25525-Breast adenocarcinoma celastrol 17252 | breast adeno-carcinoma | multiple targets | Breast | Celastrol | TRUE | 0.99966132 | 1 |
| 25083-Breast adenocarcinoma CGP60474 16761 | breast adeno-carcinoma | CDK1; CDK2 | Breast | Cgp60474 | TRUE | 0.99881416 | 1 |
| 18267-Breast adenocarcinoma AZD8055 9187 | breast adeno-carcinoma | mTOR | Breast | Azd8055 | TRUE | 0.99731849 | 1 |
| 7513-Breast adenocarcinoma OTSSP167 25473 | breast adeno-carcinoma | MELK | Breast | Otssp167 | TRUE | 0.9991679 | 1 |
| 18469-Breast adenocarcinoma HG6-64-1 9411 | breast adeno-carcinoma | B-RAF | Breast | Hg6-64-1 | TRUE | 0.99314697 | 1 |
| 25636-Breast adenocarcinoma HG6-64-1 17375 | breast adeno-carcinoma | B-RAF | Breast | Hg6-64-1 | TRUE | 0.99867832 | 1 |
| 14-Breast carcinoma estradiol 1431 | breast carcinoma | | Breast | Estradiol | TRUE | 0.99207239 | 1 |
| 895-Breast ductal carcinoma GSK1059615 27068 | breast ductal carcinoma | PI3K; mTOR | Breast | Gsk1059615 | TRUE | 0.98180702 | 1 |
| 1263-Breast ductal carcinoma lapatinib 2924 | breast ductal carcinoma | HER2; EGFR | Breast | Lapatinib | TRUE | 0.99916824 | 1 |
| 9-Normal control olive pollen extract 16317 | Normal control | | Peripheral blood | Olive pollen extract | FALSE | 0.00276633 | 0 |
| 37-Normal control 257 | Normal control | | Lung | | FALSE | 0.00027655 | 0 |
| 21926-Normal control rebastinib 13253 | Normal control | BCR-ABL | Kidney | Rebastinib | FALSE | 0.08588748 | 0 |
| 4-Normal control mock 16535 | Normal control | | Bone marrow | Mock | FALSE | 0.00030339 | 0 |
| 15884-Normal control withaferin A 6539 | Normal control | IKKβ | Breast | Withaferin A | FALSE | 0.00271459 | 0 |

**Table 2.** *Cont.*

| Analysis Name | Disease State | Target Gene | Tissue | Treatment | EMT | Prediction | Label |
|---|---|---|---|---|---|---|---|
| 4-Normal control lipopolysaccharide (LPS) 15703 | Normal control | | Embryo | Lipopoly saccharide (LPS) | FALSE | 0.00194256 | 0 |
| 10-Normal control co-culture 3076 | Normal control | | Peripheral blood | Co-culture | FALSE | 0.00115878 | 0 |
| 6-Normal control actinomycin D 4750 | Normal control | | Fetal kidney | Actinomycin D | FALSE | 0.01263976 | 0 |
| 2-Melanoma 35 | Melanoma | | Skin | | FALSE | 0.02006465 | 0 |
| 490-MYD88 deficiency lipopolysaccharide (LPS); polymyxin 12583 | MYD88 deficiency | | Peripheral blood | Lipopoly saccharide (LPS); polymyxin B | FALSE | 0.03955118 | 0 |

### 3.4. Regulation of EMT in the Development Pathway in Other Diseases Than Cancer

The results of the modeling of regulation of EMT in the development pathway found one false-positive and one false-negative result in the model Elastic-Net Classifier in the process of the model generation (Figure 5). The analysis of the false-negative result was Parkinson's disease with a z-score of 3 (Figure 5a). The analysis of the false-positive result was a genetic disease with a z-score of −2.646 (Figure 5b).

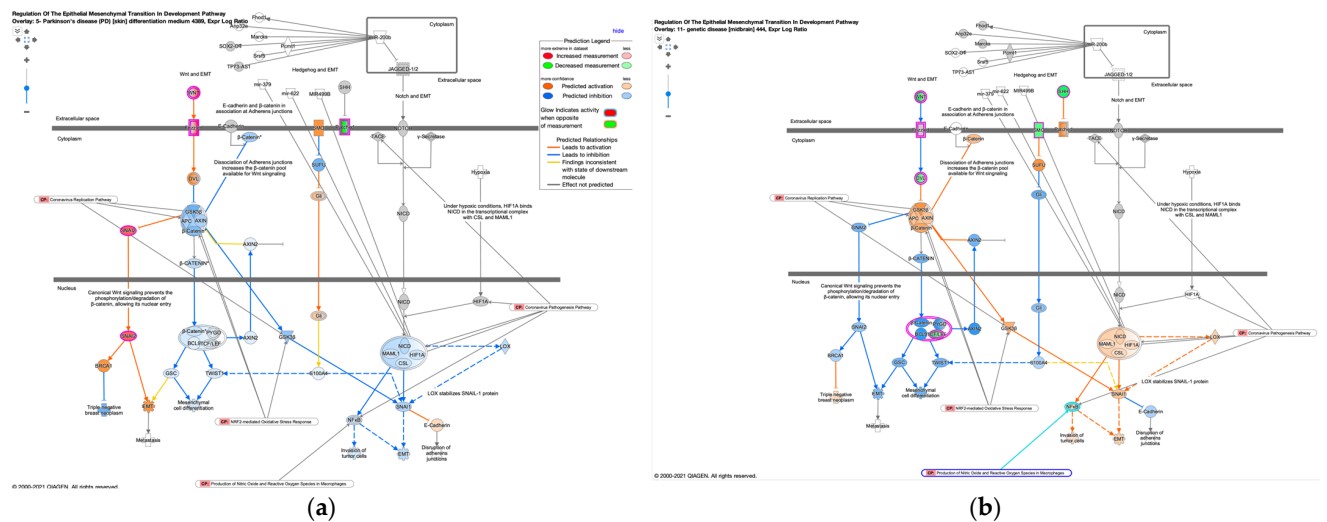

(**a**)  (**b**)

**Figure 5.** Regulation of EMT in development pathway in diseases. (**a**) Parkinson's disease (PD) (skin) differentiation medium 4389, *p* value = $1.89 \times 10^{-2}$, z-score = 3; Gene identifiers marked with an asterisk (\*) indicate that multiple identifiers in the dataset file map to a single gene in the Global Molecular Network. (**b**) genetic disease (midbrain) 444, *p* value = $4.75 \times 10^{-2}$, z-score = −2.646.

## 4. Discussion

Our result demonstrates that the canonical pathway of regulation of the EMT in the development pathway was activated in diffuse-type GC but not in intestinal-type GC. Specifically, the pathway mapping of gene expression revealed that Frizzled and APC were upregulated, while SUFU, PYGO2, and BRCA1 were downregulated in diffuse-type GC compared to intestinal-type GC. Frizzled proteins are a family of Wnt receptors involved in carcinogenesis [24]. It was previously shown that Frizzled-7 affected stemness and chemotherapeutic resistance in GC [25]. Accordingly, targeting inhibition of Frizzled-7

attenuated spheroid formation and stemness, as well as the resistance to cisplatin, an anti-cancer drug, in GC cells may have a therapeutic effect [25]. Besides Frizzled-7, the expression of Frizzled-10 was shown to have interesting correlation with cancer evolution. Importantly, as Frizzled-10 is not expressed in fully proliferative healthy tissue, but is highly expressed in certain cancerous tissue, it has the potential to be used as a prospective receptor molecule for targeted therapy. Intriguingly, it was found that while in GC, a decrease in cytoplasmic expression of Frizzled-10 is associated with increasing malignancy, while in colon cancer, the opposite is true; increased cytoplasmic expression of Frizzled-10 is crucial for the late stages of colon cancer progression and metastasis [24]. The co-localized expression of Frizzled family in different sub-types of cancer would confer progressive features on cancer.

APC is essential as a tumor suppressor protein in colorectal cancer and for its destruction complex functions, though its specific molecular activity has not been fully resolved [26]. The modeling or simulation of the cellular phenotype transition in EMT and diseases and predicting the molecular-induced responses in diseases would be useful for future investigation.

SUFU, PYGO2, and BRCA1 were downregulated in diffuse-type GC compared to intestinal-type GC. Previous findings have reported that SUFU, a regulator of Wnt signaling, was downregulated in GC and inhibited by miRNA-324-5p [27]. It was suggested that miRNA-324-5p induces EMT by inhibiting SUFU in GC [27]. PYGO2 was reported to be increased in human breast cancer [28]. The expression of PYGO2 was also assessed in glioma tissue samples and the results showed a positive correlation between tumor grade and PYGO2 overexpression [29]. The expression of PYGO2 was overexpressed in drug-resistant cell lines of GC and GC tissue after neoadjuvant chemotherapy [30]. It may be possible that PYGO2 has a different expression profile in diffuse-type GC compared to intestinal-type GC. BRCA1 was also downregulated in diffuse-type GC compared to intestinal-type GC. We have previously shown that the role of BRCA1 in the DNA damage response pathway was activated in intestinal-type GC compared to diffuse-type GC [18]. Accordingly, BRCA1 is rather important to intestinal-type GC.

The current study successfully generated AI-based models using 50 activated and 50 inactivated images of EMT gene regulation in the development pathway. The analyses in the database were selected based on the diseases and the treatment (Tables 1 and 2). Diseases in activated states of EMT regulation in the development pathway included bone osteosarcoma [31], breast carcinoma [32], and colon cancer [33]. AI application in gastrointestinal diseases would be a promising approach [34].

An interesting point of our current study is that the machine-learning modeling revealed that an IPA analysis of Parkinson's disease had a false-negative prediction result (Figure 5a). The color of the picture seems to be inactivated, which is in accordance with the prediction result as inactivated. Furthermore, it seems that EMT activation in the WNT pathway via SNAI2 resulted in the prediction being activated, whereas CSL-HIF1A-MAML1-NICD complex-induced EMT via SNAI1 was predicted as inactivated. In addition to Parkinson's disease, the machine-learning modeling revealed that an analysis of another unrelated genetic disease had a false-positive prediction result (Figure 5b). On the other hand, based on the analysis, GSK3β and SNAI1 were predicted as activated, while SNAI2 was inactivated (Figure 5b). The activation of GSK3β could be associated with the mediator role of GSK3β in the cross-talk of EMT signaling pathways [35].

## 5. Conclusions

The regulation of EMT in the development pathway was activated in diffuse-type GC and inactivated in intestinal-type GC. AI modeling with molecular pathway images generated the Elastic-Net Classifier model. The validation with 10 activated and 10 inactivated new pathway images, which were not used for the modeling, resulted in high accuracy. The modeling of the cellular phenotype transition in EMT and diseases will be studied in the near future.

**Author Contributions:** Conceptualization, S.T.; methodology, S.T.; formal analysis, S.T.; investigation, S.T.; writing—original draft preparation, S.T.; writing—review and editing, S.T., S.Q., R.O., H.C., K.A., A.H., E.J.P., H.Y. and H.S.; visualization, S.T.; project administration, S.T.; funding acquisition, S.T., S.Q., R.O. and A.H. All authors have read and agreed to the published version of the manuscript.

**Funding:** This research was funded by the Japan Agency for Medical Research and Development (AMED) Grant Number JP20ak0101093 (S.T., R.O. and A.H.), JP21mk0101216 (S.T.), JP22mk0101216 (S.T.), and Strategic International Collaborative Research Program, Grant Number JP20jm0210059 (S.T. and S.Q.), Japan Society for the Promotion of Science (JSPS) KAKENHI Grant Number 21K12133 (S.T. and R.O.).

**Institutional Review Board Statement:** Not applicable.

**Informed Consent Statement:** Not applicable.

**Data Availability Statement:** Not applicable.

**Acknowledgments:** The authors would like to acknowledge Shinpei Ijichi for assisting with the DataRobot Automated Machine Learning platform. The authors are grateful to all colleagues including members of the National Institute of Health Sciences, Japan for their support. This research was supported by the Ministry of Health, Labour, and Welfare, Japan.

**Conflicts of Interest:** The authors declare no conflict of interest.

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
