# Peer review of "Regulation of Epithelial–Mesenchymal Transition Pathway and Artificial Intelligence-Based Modeling for Pathway Activity Prediction"

_onco, doi:10.3390/onco3010002_

Round 1
Reviewer 1 Report
This is an interesting manuscript in which the authors have used Artificial Intelligence (AI)-based modeling to predict pathways perturbance in epithelia-mesenchymal transition underlying diffuse-type gastric cancer.
The AI-based bioinformatics and the derived dats are quite strong. However, the lack of experimental validation of the in silico analysis is a concern. Validation using a different algorithm (Table 1) is often deemed less significant compared to experimental data. The manuscript can be greatly strengthened by the addition of in vitro or in vivo experimental data from representative activated and inactivated pathway analysis.
Author Response
Thank you very much for your kind comments. This study focuses on in silico analysis, the in vitro or in vivo experiment would be investigated from representative activated and inactivated pathway analysis in the future.

Reviewer 2 Report
Please see attached file for specific comments
This manuscript reports an interesting topic in cancer, Epithelial–Mesenchymal Transition Pathway, and developing the AI model for prediction pathway activity. However, the manuscript lacks the connection between different result parts and is not really meaningful. It has to be improved and the resolution of
the figure too low, can not see what is inside.
Following are specific comments
Major comments
The result lacks the Different expression gene analysis between Diffuse- and Intestinal-Type GC based on the dataset
Figures 1, 2, 5. The resolution of this figure is too low, and the text size is too small, can not read P5, part 3.3 . The AI modeling and methods were not clear. What is the expected outcome? The current results didn’t show clearly the regulation of EMT.
P5, part 3.3. the diseases listed in table 1 have not been mentioned before. Should it mention in the method section as data collection, and why chose the disease data?
The discussion part did not really support the results, especially the EMT AI modeling and results.
Minor comments
P2, line 59-62: the previous sentence mention EMT in drug resistance but the rest does not address thatP2, line 63. Suddenly mention silibinin, should remove this information.
P2, part 2.1, should indicate the specific name of the TCGA dataset was used
P3, line 121: “(Entrez Gene Summary, Gene ID: 324)” should write at the first time mention APC geneP7, line 199-201, suddenly mention methylation and l ong non-coding RNA. There is no connection with other information.
Author Response
Thank you very much for your thoughtful and concrete comments.
Major comments
The result lacks the Different expression gene analysis between Diffuse- and Intestinal-Type GC based on the dataset
Reply: the different expression gene analysis has been published in our previous study: Cancers 2020, 12(12), 3833; https://doi.org/10.3390/cancers12123833
We added the description on the gene expression analysis in Results (line 440-442):
“Alteration in gene expression in diffuse- and intestinal-type GC was mapped in a canonical pathway, “Regulation of the EMT in development pathway” (Figure 1) based on the previous gene expression analysis result [1].”Figures 1, 2, 5. The resolution of this figure is too low, and the text size is too small, can not read P5, part 3.3 . The AI modeling and methods were not clear. What is the expected outcome? The current results didn’t show clearly the regulation of EMT.
Reply: The figures were replaced figures in high-resolution as possible. Part 3.3 AI modeling and validation of the prediction model was revised to indicate that the activation map highlighted the parts of the image data critical to the prediction accuracy of the model in an activation map (Figure 4) (lines 497-503):
The activation state of regulation of EMT in the development pathway was modeled by machine learning, including deep learning, using 50 activated and 50 inactivated images of the regulation of EMT in development pathway (Figure 4). DataRobot was used for machine-learning modeling and 34 models were automatically created, including an Elastic-Net Classifier (L2/Binomial Deviance) model. DataRobot also highlighted the parts of the image data critical to the prediction accuracy of the model in an activation map (Figure 4).
P5, part 3.3. the diseases listed in table 1 have not been mentioned before. Should it mention in the method section as data collection, and why chose the disease data?
The discussion part did not really support the results, especially the EMT AI modeling and results.
Reply: The disease list based on database search in IPA is summarized in new Table 1 in the method section. The description was added in the method section (lines 244-249):
The description of the EMT AI modeling was added in discussion (lines 574-578).Minor comments
P2, line 59-62: the previous sentence mention EMT in drug resistance but the rest does not address thatP2, line 63. Suddenly mention silibinin, should remove this information.
Reply: Description of silibinin was deleted in around line 219.
P2, part 2.1, should indicate the specific name of the TCGA dataset was used
Reply: the specific name of the dataset was added in lines 235-236.P3, line 121: “(Entrez Gene Summary, Gene ID: 324)” should write at the first time mention APC gene
Reply: It was deleted to avoid any confusion.
P7, line 199-201, suddenly mention methylation and long non-coding RNA. There is no connection with other information.
Reply: Deleted.
